# Physical and Mechanical Properties of High-Density Fiberboard Bonded with Bio-Based Adhesives

Aneta Gumowska 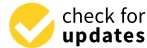 and Grzegorz Kowaluk *

Institute of Wood Sciences and Furniture, Warsaw University of Life Sciences—SGGW, 02-776 Warsaw, Poland
* Correspondence: grzegorz_kowaluk@sggw.edu.pl

**Abstract:** The high demand for wood-based composites generates a greater use of wood adhesives. The current industrial challenge is to develop modified synthetic adhesives to remove harmful formaldehyde, and to test natural adhesives. The scope of the current research included the manufacturing of high-density fiberboards (HDF) using natural binders such as polylactic acid (PLA), polycaprolactone (PCL), and thermoplastic starch (TPS) with different resination (12%, 15%, 20%). The HDF with biopolymers was compared to a reference HDF, manufactured following the example of industrial technology, with commonly used adhesives such as urea-formaldehyde (UF) resin. Different mechanical and physical properties were determined, namely modulus of rupture (MOR), modulus of elasticity (MOE), internal bonding strength (IB), thickness swelling (TS), water absorption (WA), surface water absorption (SWA), contact angle, as well as density profile; scanning electron microscope (SEM) analysis was also performed. The results showed that increasing the binder content significantly improved the mechanical properties of the panels in the case of starch binder (MOR from 31.35 N mm$^{-2}$ to 40.10 N mm$^{-2}$, IB from 0.24 N mm$^{-2}$ to 0.39 N mm$^{-2}$ for dry starch), and reduces these in the case of PLA and PCL. The wet method of starch addition improved the mechanical properties of panels; however, it negatively influenced the reaction of the panels to water (WA 90.3% for dry starch and 105.9% for wet starch after 24 h soaking). Due to dynamically evaporating solvents from the PLA and PCL binding mixtures, a development of the fibers' resination (blending) techniques should be performed, to avoid the uneven spreading of the binder over the resinated material.

**Keywords:** high density fiberboard; wood; mechanical properties; physical properties; polylactide; polycaprolactone; thermoplastic starch

## 1. Introduction

Progressive economic development and the growing population significantly affect both the natural environment and the global biomass resources, including wood, forestry, and agricultural by-products, based on which wood-based panels are produced. Wood-based panels can be classified into three basic groups depending on the form of their structural elements. The structure and technological process determine the mechanical and physical properties. The first group consists of fiber composites, including boards manufactured using the dry forming method, where the fiber-carrying medium is air; this includes high-density fiberboards (HDF), medium-density fiberboards (MDF); and low-density fiberboards (LDF). The wet forming method, where the medium is water, is used to produce softboards (SB) and hardboards (HB). Fiber composites also include wood-plastic composites (WPC). The second group consists of particleboard composites, such as particleboards (PB) and oriented strand boards (OSB). The third group is plywood, and laminated veneer lumber (LVL), representatives of layered composites. The production of wood-based panels increases from year to year, reaching approx. 167 million m$^3$ for plywood, approx. 110 million m$^3$ for fiberboards, and approx. 100 million m$^3$ for particleboards [1,2]. The high demand for wood-based panels generates a greater use of wood ad-

hesives, which automatically translates into the use of synthetic adhesives. Formaldehyde-based adhesives account for approx. 90%–95% of all wood adhesives used in the industry, and urea formaldehyde (UF) is the leader in terms of consumption, accounting for approx. 11 million tons of resin dry weight per year [3,4]. The global wood adhesives and binders market size reached 15.8 billion US dollars (USD) in 2020, and it is projected to grow by approx. 4% to reach 21.9 billion USD by 2028 [5]. It is worth adding that the above data include UF, phenol-formaldehyde (PF) resins, and soy-based binders, the latter being a good indicator for new green adhesives.

Until the end of the 1950s, adhesives of natural origin were available, which had defects, e.g., low water resistance and durability, resulting in the development of petro-chemical adhesives. Synthetic formaldehyde-based adhesives have gained popularity and lack competition from natural adhesives, due to their low production cost while main-taining perfect adhesive properties such as excellent stability, low curing temperature, and fast curing, colorless and resistance to microorganisms [6,7]. The biggest challenge for formaldehyde-based resin technology is to reduce the emission of free formaldehyde, which is harmful to health and the environment during production and in final products during their entire life cycle. While levels are generally low, increasingly stringent environmental regulations will require more environmentally friendly adhesives to produce bio-panels [8]. The turn of the 20th and 21st centuries is a period of return to the development of natural adhesives, the desire to look for an alternative to conventional petroleum-based adhesives by reducing or eliminating formaldehyde, and the sustainable development of raw materials and final products [9].

MDF is a wood-based board produced by pressing lignocellulose fibers together with synthetic resins under conditions of high temperature and pressure [10]. Wood fibers comprise the dispersed phase, which is responsible for the strength and stiffness of the board, while the matrix used is a construction binder, and can be a synthetic resin, an inorganic compound, or a biopolymer. [11]. Currently, commonly used resins in the production of MDF panels are UF, melamine urea formaldehyde (MUF), PF, and isocyanate compounds [12]. On a global production scale, 90% of MDF panels are made with UF resin [6]. Commercial MDF has a higher density and strength than particleboard or plywood. Unlike particleboard, which is also commonly and widely used in furniture and construction, the MDF board has a homogeneous structure over the entire cross-section; because of this, it has excellent mechanical workability in the milling process, a high precision finish which provides an excellent base for veneers, and high dimensional stability, making it ideal for use in building materials or furniture boards [13–15].

With the predominance of petrochemical-based products raising concerns about limited fossil resources and harm to health and the environment, it is becoming prudent to obtain materials that will have similar properties to the properties of commercial wood-based panels with one difference—they will come from renewable raw materials.

A future-proof alternative, and at the same time a response to the restrictions and needs of the market, is the production of bio-composites, which means that one of at least two component materials will be of natural origin. Recyclable composites are being developed more widely due to the possible use of renewable raw materials: by-products from the agricultural industry, such as sugarcane bagasse [16], kenaf stalks [17], rice husk [18], corn [19], tomato stalks [20], sugar beet pulp [21], walnut shell [22], and forest by-products such as cones [23,24], tree bark [25,26], and branches [27]. The current state-of-the-art of particleboard manufacturing which uses environmentally-friendly agricultural biomass was widely reviewed and described by researchers [28,29]. Another important aspect of bio-composites is natural and renewably sourced adhesives. Synthetic resin can be replaced with natural binders, i.a. starch and its modifications [30–34] lignin [35], cellulose [36], soybean and soybean protein [37–40], chitosan [41], tannins [42,43], vegetable oils [44], natural rubber [45,46], and citric acid [47,48]. The use of biopolymers as a binder in wood-based material technologies has also been extensively studied. Biopolymers such as polylactic acid (PLA), polyhydroxybutyrate (PHB), and polycaprolactone (PCL) are also

used as alternatives to synthetic adhesives. PLA is the most popular and the only biopolymer currently produced on an industrial scale. The production scale was 150,000 metric tons in 2013, with an increase of 10% in 2020. Currently, the production process of PLA generates 60% fewer greenhouse gases and uses 50% less non-renewable energy than the production of traditional polymers such as polyethylene terephthalate (PET) or polystyrene (PS) [49].

Song et al. [50] manufactured board in a wet method process using soybean straw fiber, with varying amounts of PLA as a binder (0, 10, 30, and 50%). They proved that the addition of PLA weakened the mechanical properties of tested boards. However, when it came to hygroscopic properties, they showed that increasing the amount of PLA improved the water resistance. Ultimately, they confirmed the validity of using PLA as a binder for fiberboards; however, a coupling agent was needed to improve the mechanical properties. Ye et al. [51] produced fiberboards from softwood fiber, wheat, and soybean straws. They proved that fibers from wheat and soybean straws can be used in a 50/50% agricultural fiber/wood fiber ratio, because they could provide comparable mechanical properties and water resistance to fiberboards made of 100% wood fiber. Many literature reports on PLA, PCL, and PHB concerning fiber composites refer to wood-plastic composites (WPC), produced by methods such as direct injection molding, extrusion, compression molding, film and sheet formation [52–56].

The current state of the art is limited on the subject of using biopolymers, e.g., PLA, PCL, PHB, as a binder in wood-based panel technology, particularly with medium-density fiberboards. This investigation aimed to assess the impact of natural biopolymer binders—thermoplastic starch (TPS), PLA, and PCL—on selected physical and mechanical properties of HDF manufactured with different resination.

## 2. Materials and Methods

### 2.1. Materials and Their Characterization

In the present study, high-density fiberboards (HDF) were produced under laboratory conditions from pine (*Pinus sylvestris* L.) and spruce (*Picea abies* L.) industrial fibers (IKEA Industry Poland sp. z o. o. brand Orla, Szczecin, Poland). The fibers were dried to a moisture content (MC) of about 4%.

Composites were manufactured with four different binders: pure, laboratory-purpose polylactide (PLA; Sigma-Aldrich, Saint Louis, MO, USA, product no. 38534), polycaprolactone (PCL; Sigma-Aldrich, product no. 704105) in drops with a diameter of 3 mm, thermoplastic starch (TPS; Grupa Azoty S.A., Tarnów, Poland) in drops and powder form, and urea-formaldehyde resin (UF; Silekol S-123, Silekol Sp. z o.o., Kędzierzyn-Koźle, Poland), the latter of which is commonly used in industry.

### 2.2. Fiberboard Manufacturing

The HDF were manufactured with dimensions 320 mm × 320 mm, a target density of 900 kg m$^{-3}$, and an average thickness of 3 mm. The fiberboards were produced using different binders as well as different resination values. All composites with biopolymers as a binder were manufactured with 12%, 15%, and 20% resination. The reference variant was produced with 12% resination, as in industrial conditions. The resination values used in this study were previously used for particleboard with biopolymers as an adhesive [57]. Increasing the resination from 12% to 15% significantly improved the physical and mechanical properties of the produced particleboards. However, increasing from 15% to 20% seemed unjustified. Therefore, in this study, the same resination values were used, but not exceeding 20%. No paraffin emulsion or wax was added during the manufacturing of the high-density fiberboards. The binder liquid solutions were mixed in a laboratory mixer and sprayed by air gun onto the lignocellulosic fibers. The TPS powder was manually applied into fibers, and then water was sprayed using a spray gun, with the amount of water depending on the percentage of resination. After applying biopolymers and blending, the resinated fibers were stored in the laboratory fume hood for 3 days to evaporate

the solvents. Then, the mats were formed and pre-densified manually. To improve the heat transfer into the core of HDF, 65 g m$^{-2}$ of water was sprayed on the top and bottom sides before hot pressing. The HDF was pressed in two stages: cold pressed (unit pressure 0.8 MPa, 30 s) to thicken the mat, and then hot pressed for 10 min at 180 °C under 2.5 MPa unit pressure (AKE, Mariannelund, Sweden). For the reference panels, the hot pressing time was reduced to 60 s, while other pressing parameters remained unchanged. For the biopolymers, an extended pressing time was used to allow the thermoplastic binders to properly melt. In the process of manufacturing wood-based panels, the pressing temperature depends on the binder used. DSC and TGA analysis for PLA, PCL [58], and starch [59] allowed us to determine the "safe" temperature of 180 °C, which does not cause degradation of the biopolymer and is sufficient to melt.

As a result, thirteen types of fiberboards were produced with different resination (hereby referred to by the shortcuts listed in Table 1), and two panels of each binder type. The manufactured HDF were conditioned in ambient conditions (20 °C; 65% R.H.) until they reached a constant weight, before being cut according to the research schedule.

**Table 1.** Shortcuts for every elaborated HDF.

| Variants of Binders and Resination | Shortcut of HDF |
|---|---|
| UF 12% | UF12 |
| Powder TPS 12%—dry starch | DS12 |
| Powder TPS 15%—dry starch | DS15 |
| Powder TPS 20%—dry starch | DS20 |
| Drops of TPS 12%—wet starch | WS12 |
| Drops of TPS 15%—wet starch | WS15 |
| Drops of TPS 20%—wet starch | WS20 |
| PLA12% | PLA12 |
| PLA15% | PLA15 |
| PLA20% | PLA20 |
| PCL12% | PCL12 |
| PCL15% | PCL15 |
| PCL20% | PCL20 |

### 2.3. Preparation of the Adhesives

The adhesive mass for PLA, PCL, and TPS (drops) was produced by dissolving the dry mass of polymers in solvent to obtain the consistency of a thick liquid. The following solvents have been used to achieve liquid-state binders: methylene chloride ($CH_2Cl_2$) for PLA (about 23% dry matter content of resulting mixture), toluene ($C_7H_8$) for PCL [57] (about 27% dry matter content of resulting mixture), and hot water for TPS (drops). The TPS drops with hot water were mixed using a low-speed mixer, until a homogeneous suspension was achieved (about 25% dry matter content). All solution concentrations were tuned to obtain the viscosity of solutions similar to the reference binder (in the range of 420–450 mPa s). The reference HDF was produced with UF industrial resin (65% dry matter content), where the hardener was a 10% water solution of ammonium sulfate (($NH_4$)$_2SO_4$) in a weight ratio of 50:5:1.5, respectively: resin: water: hardener. The curing time of the reference bonding mixture at 100 °C was about 82 s.

### 2.4. Physical and Mechanical Properties

The modulus of rupture (MOR) and modulus of elasticity (MOE) were characterized according to EN 310 [60], and the internal bond (IB) was determined according to EN 319 [61]. The mechanical properties were analyzed with an INSTRON 3369 (Instron, Nor-

wood, MA, USA) standard laboratory testing machine, and as many as 10 test specimens for each fiberboard type were analyzed for the tests mentioned. Thickness swelling (TS) and water absorption (WA) at two-time intervals, i.e., after 2 h and 24 h of immersion in water, were measured according to EN 317 [62], and 12 samples of HDF panels of each binder variant were used. Surface water absorption was conducted according to EN 382–2 [63], in two repetitions for each variant of the panels. The density profile (DP) of samples was analyzed using a DA-X measuring instrument (GreCon, Alfeld, Germany). The measurement based on direct scanning X-ray densitometry was carried out at a speed of 0.05 mm s$^{-1}$ across the panel thickness with a sampling step of 0.02 mm. The density profile was performed using three samples of each variant for a type of binder.

Contact angle measurements were made using the contact angle analyzer PHOENIX 300 (SEO—Surface & Electro Optics Co., Gyeonggi-do, Ltd., Suwon City, Korea) equipment, using the method of distilled water sessile drop. A Quanta 200 (FEI, Hillsboro, OR, USA) scanning electron microscope was used to define the surface morphology of the produced fiberboards.

### 2.5. Statistical Analysis

Analysis of variance (ANOVA) and t-test calculations were used to test for significant differences ($\alpha = 0.05$) between factors and levels where appropriate, using the IBM SPSS statistic base (IBM, SPSS 20, Armonk, NY, USA). A comparison of the means was performed when the ANOVA indicated a significant difference, by employing the Duncan test. The detailed *p*-values have been attached as Supplementary Material.

### 3. Results and Discussion

The density profile was characterized, and the results are summarized in Figure 1. The average densities of all samples are ~900 kg m$^{-3}$. The density profiles for individual samples of HDF were symmetrical to the middle of the thickness of the board; therefore, the graph presents the density profiles to their axis of symmetry to facilitate the analysis. Regardless of the biopolymer type or the resination used, the shape of all profiles did not differ significantly. Each of the profiles presented in the graph shows a characteristic density profile of HDF, where the core layer has a slightly lower density than the surface layers [64]. The boards produced were 3 mm thick; therefore, the difference between the surface layer density (SLD) and core density (CD) is smaller than in the case of 10 mm thick MDF boards, where the density distribution shows clear differences between the layers [65]. The effect of fiber compression is more even over the entire cross-section of the thinner board, as the set temperature of the press shelves reaches the core layer faster, causing hardening or binder liquefaction depending on the binder used [66]. The average density of the PCL12 samples on the surface achieved a value of 1090 kg m$^{-3}$ and recorded a decrease in density in the core layer by 24%. The most homogeneous density profile was recorded for the PLA20 samples, where density decreases in the core layer by about 6%. The HDF was initially cold-pressed to densify the mat and then hot-pressed to achieve a more even density distribution. However, it should be pointed out that the mats were sprayed with water on the surfaces to improve the heat transfer during hot-pressing. This promotes a higher densification of the face layers of the produced panels with biopolymers. Wong et al. [66] manufactured 12 mm MDF which was subjected to a two-step hot pressing process. They showed that the final profile is significantly dependent on the thickness achieved in the first stage of pressing. Obtaining a smaller thickness in the first stage resulted in a greater difference between maximum surface layer density (SLD) and core density (CD). There is no noticeable difference between the profiles for thermoplastic polymers and the UF resin used in the reference panel. Figure 2 shows the changes in the appearance of the surface of the produced HDF, depending on the binder used and the resination. HDF with PLA and PCL as a binder has visible adhesive stains, which were caused by the too fast evaporation of the solvent during the covering of the fibers. Stains indicated that the biopolymer did not evenly cover the fibers in the entire board. In the boards there are places with a

large amount of biopolymer, and there are also places where this biopolymer has been minimally distributed on the fibers. The graph shows that for PLA and PCL, the density profile is more homogeneous with increasing of resination.

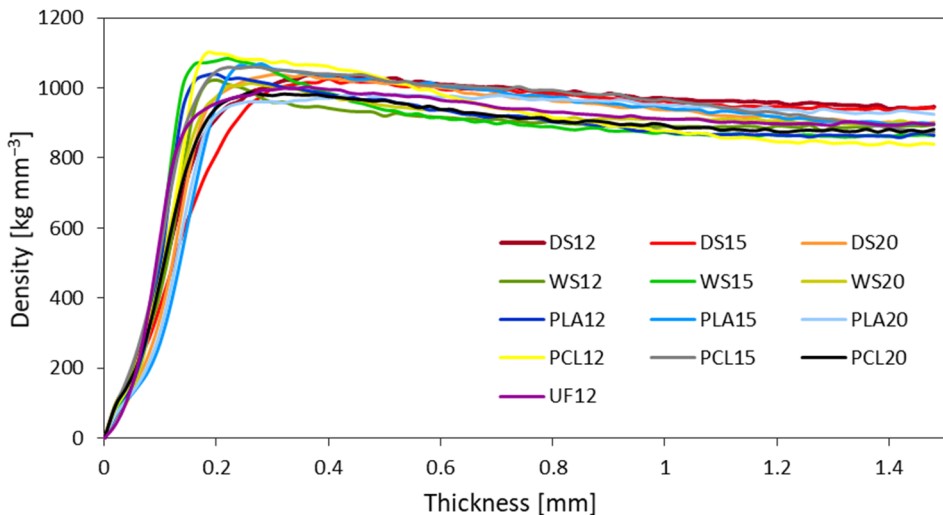

**Figure 1.** Density profiles of fiberboards.

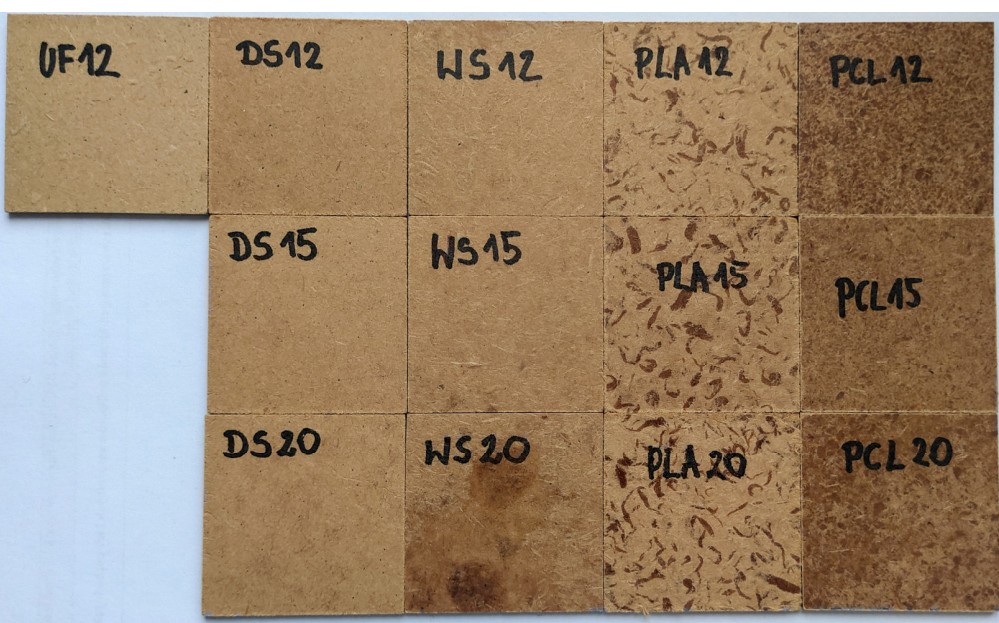

**Figure 2.** Changes in the appearance of the surface of the produced HDF, depending on the binder used and the resination.

The average values of modulus of rupture (MOR) and modulus of elasticity (MOE) under the three-point bending stress of the tested composites are presented in Figure 3. The error bars in the graphs represent the standard deviation. The highest value of MOR (59.18 N mm$^{-2}$) was recorded for the reference HDF, while the lowest (23.64 N mm$^{-2}$) was for PLA20. In the case of MOE, the highest average value was obtained for PS20 (4225 N mm$^{-2}$), and the lowest was for PCL12 (2596 N mm$^{-2}$), as well as for MOR. Analyzing the obtained results for DS, it can be observed that with the increase in resin content from 12, 15 to 20%, the average value of both MOR and MOE increases. For the WS samples, an increase in the MOR and MOE values can be seen with an increase in the resin content from 12 to 15%. The addition of 20% starch (WS) caused a decrease in strength and MOE. In the case of PLA and PCL, we can observe the opposite situation as in the case

of DS12-DS20 boards; the average values of MOR and MOR decrease with the increase in resination. It can also be seen that between 12–15% differences in the average MOR and MOE values for PLA and PCL are smaller than at 15–20%. Statistically significant differences among all variants of panels with PLA and PCL can be noted between variants PCL15 and PCL20. The explanation for the decrease in the MOR and MOE values for PLA and PCL with increasing resin content can be found in the structure of the produced HDF. As already shown above in Figure 2, there was a problem with the uniformity of the distribution of biopolymers on the fibers. The WS20 samples also show a decrease in the MOR and MOE values, although they increased from WS12 to WS15. The photo shows surface defects; thus, we can predict what the structure looks like in a cross-section of the samples. The more biopolymer, the more stain and weak areas in the board. Xiaowen et al. [50] confirmed that the addition of PLA as a binder weakened the mechanical properties (bending strength and tensile strength) of fiberboard produced using a wet method, using soybean straw fiber. They used different pressure values, which affected the final results of the tests. Of the two forms (powder, drops) of applying TPS to the fibers, the starch drops in the form of a suspension show better strength properties. The MOR value for WS12 relative to DS12 and WS15 to DS15 is higher by 32% and 30%, respectively. There are no statistically significant differences between DS20 and WS20. According to statistical analysis, there are statistically significant differences between the average values of the MOR results for UF12 and the rest of the HDF variants, also between the DS, WS, PLA, and PCL20 samples. Taking into account statistical analysis, there are no statistically significant differences between the average values of the MOE results from UF12, WS15, and WS20; DS12-DS20, WS20, PLA12, PLA15, and PCL12.

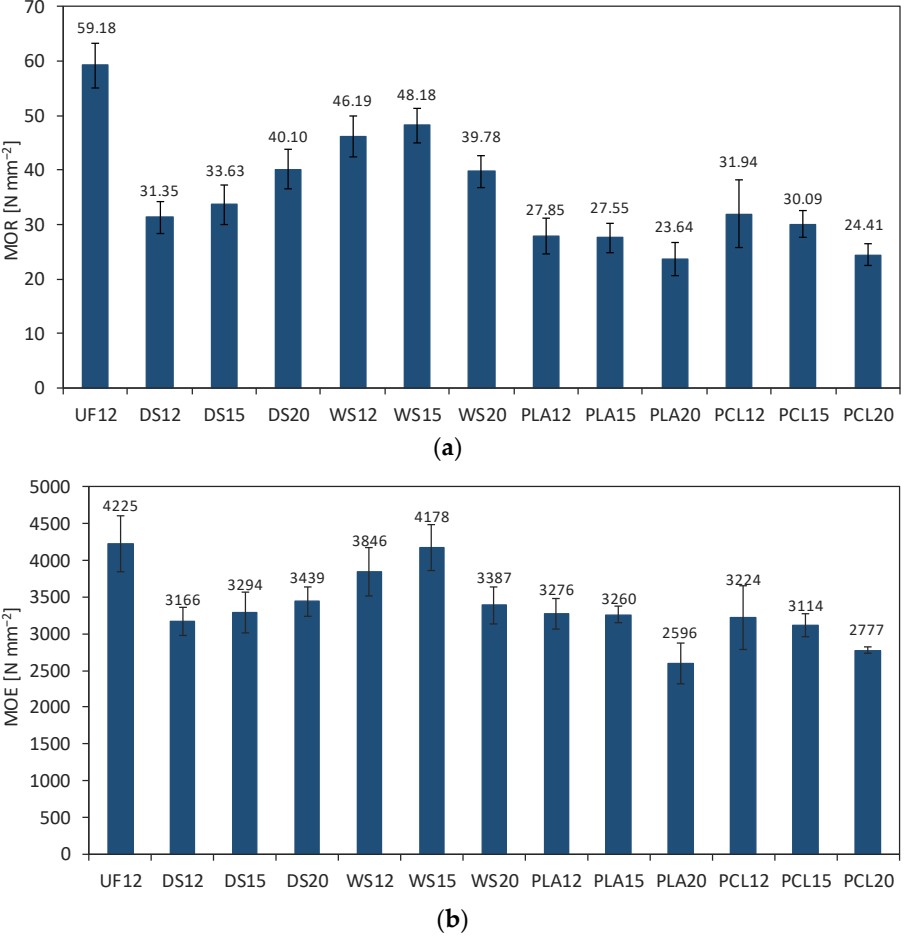

**Figure 3.** (**a**) Modulus of rupture (MOR) and (**b**) modulus of elasticity (MOE) of tested HDF.

The obtained results of internal bonding (IB) tests are presented in Figure 4. The error bars in the graph represent the standard deviation. The outcomes show that the highest average value of IB was that of the reference HDF (1.73 N mm$^{-2}$), and the lowest value was for DS12 (0.24 N mm$^{-2}$). Analyzing the results, it can be seen that the lowest IB values were recorded for DS and PLA samples, regardless of the level of resination. The same relation refers to IB as to MOR and MOE. For DS, an increase of resination from 15% to 20%, and for WS12 to WS15, causes a slight increase in average IB values. The PLA and PCL board variants show a decrease in the IB with an increase in resination. Better results in this test were also obtained by starch in the form of drops, which were applied to the fibers in the form of a thick liquid using a pneumatic gun. This method of the application allowed us to obtain better coverage of the fibers. The starch powder probably did not fully mix with the introduced water, leaving zones where the binder and fibers did not mix well. No significant influence of the density profile on the IB has been found. According to statistical analysis, statistically significant differences exist between the average values of IB results for the reference HDF and the rest of the panels produced; between DS12-DS20, PLA12-PLA20, and variants WS12-WS20, PCL12-PCL20. The representative forms of damage of the samples resulting after the IB test are compiled in Figure 5. The first group of destruction took place in the middle of the composite thickness. The first form of destruction was obtained by UF12, WS12, WS15, WS20, PCL12, PCL15, and PCL20. The second group includes samples in which the destruction took place in the near-surface zone; it includes samples with an average IB below 0.48 N mm$^{-2}$. The HDF with damage in the near-surface layers had the weakest strength of all tested samples, which is caused by more effective bonding in the core layer. Ji et al. [41] developed a method for manufacturing medium-density fiberboard (MDF) adhesives using chitosan as the main component and glutaraldehyde as a crosslinking agent. They prove that an increasing amount of glutaraldehyde could be contributing toward the deterioration of the mechanical properties of MDF. The optimal IB value was 1.22 N mm$^{-2}$, with the board sanded to reduce the low-density surface area which results in an improved surface quality and finish.

Baskaran et al. (2012) [67] conducted tests with the addition of freeze-dried and pure polyhydroxyalkanoates (PHA) in fine particles of oil palm trunk at different amounts. They proved that MOR and MOE increased as the amount of both forms of PHA increased.

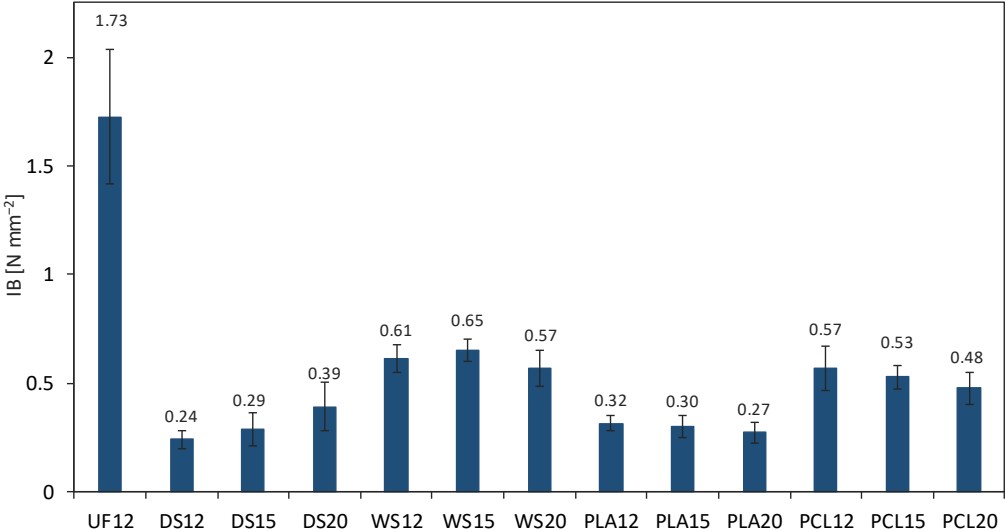

**Figure 4.** Internal bond (IB) of tested HDF.

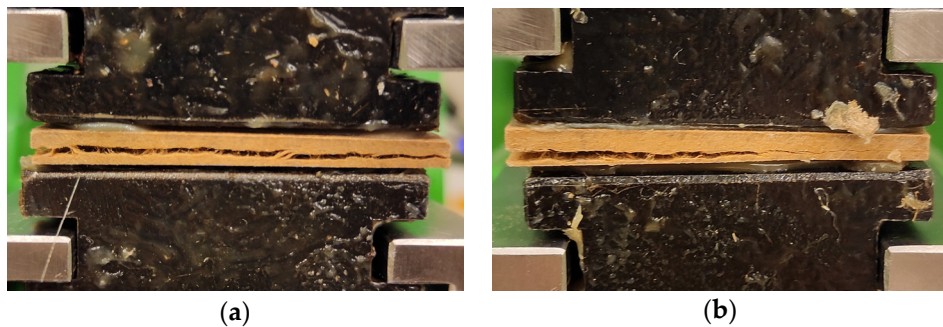

|  (**a**)  |  (**b**)  |
|---|---|

**Figure 5.** Two representative forms of damage after the IB test (**a**) in the core layer; (**b**) near-surface zone.

Scanning electron microscopy (SEM) was used to analyze the bonding quality of HDF with biopolymers and UF resin directly at the microstructure scale. The photos in Figure 6 are presented for each of the produced variants of panels, and were marked with arrows to exemplify the characteristic areas that clearly show the wood fiber covered by the binder used, and in the case of DS samples, how powder grains are visible. There is a visible increase in the number of biopolymers covering the fibers with increasing resination. The SEM pictures show the difference between the bonding of PLA and PCL with fibers, and the reference board based on UF resin. The binder was not distributed uniformly, and places where the fibers are fully covered by the biopolymer were observed in HDF with PLA and PCL, which potentially affected the mechanical and physical properties. SEM photos of PLA-based composites with the addition of agricultural by-product fibers show poor fiber-to-matrix adhesion, voids, and fiber breakage [68]. Song et al. [50], performed an SEM analysis, proving that the addition of 10% PLA in a fiberboard covers insignificant part fibers, which causes gaps between the fibers. The addition of 30–50% PLA causes the penetration of PLA into the gaps between fibers, which has a positive effect on water resistance and dimensional stability. Sivakumar et al. [69] observed an even distribution of banana leaf fiber with thermoplastic cassava starch as a strong adhesive matrix, which improved the mechanical properties of the bio-composites. The composite–water interaction (like TS, WA, and SWA) can be dependent on the tested panels' microstructure and binder distribution.

The average values of thickness swelling (TS) and water absorption (WA) on HDF after 2 h and 24 h are presented in Figure 7a–b. The error bars in the graphs represent the standard deviation. The average TS of the specimens after 2 h of immersion ranged from 10.4% for UF12 to 37.6% for DS12. After 24 h of immersion, the results were between 21.7% and 80.4% for UF12 and WS12, respectively.

Under industrial conditions, for the production of MDF, about 1% of paraffin emulsion is added to ensure water resistance; in contrast, no hydrophobic agents were used in these studies. The reference composites use UF resin, which is intended to be used in dry conditions. For each tested group of biopolymers, it can be observed that TS decreases with increasing resination. Biopolymers act as a binder but also fill the voids between the fibers, acting in favor of water resistance. Increasing the resination was the least effective for HDF with PLA as a binder, both after 2 h and 24 h of soaking in water. The SEM analysis confirmed that the increase in resination visibly shows a greater fiber surface coverage by the biopolymer, which translates into a higher water resistance. The results of water absorption (WA) of the tested HDF after 2 h and 24 h of immersion in water show a trend consistent with the results of TS. The lowest value of WA either after 2 h and 24 h was achieved for the reference HDF, while the highest obtained after 2 h was DS12 and after 24 h was WS12.

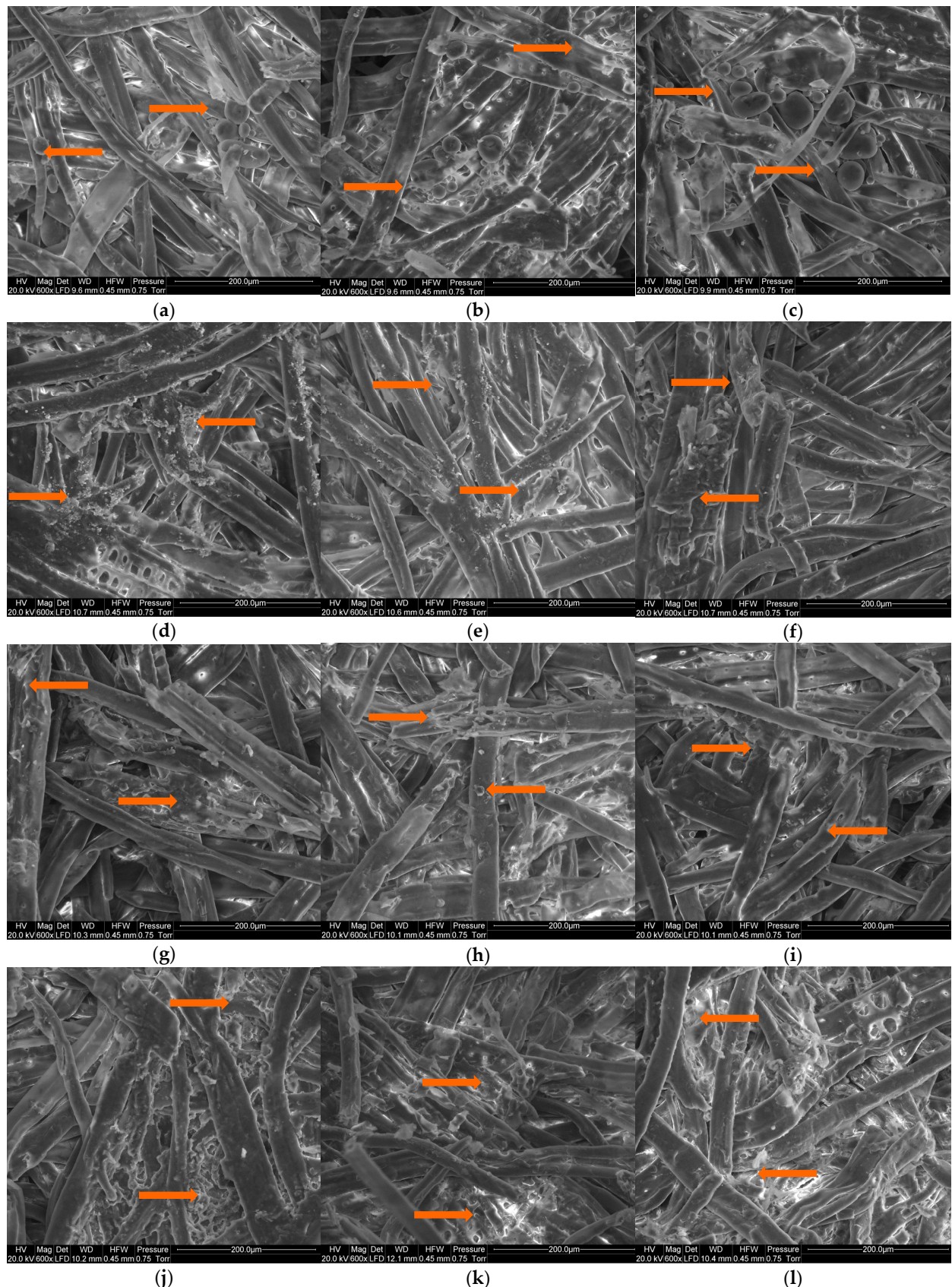

**Figure 6.** *Cont.*

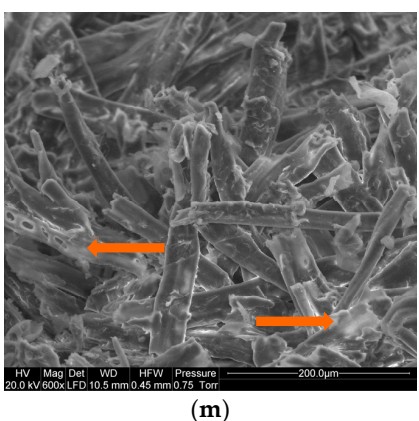

(**m**)

**Figure 6.** Scanning electron microscope images of HDF with, (**a**) DS12, (**b**) DS15, (**c**) DS20, (**d**) WS12, (**e**) WS15, (**f**) WS20, (**g**) PLA12, (**h**) PLA15, (**i**) PLA20, (**j**) PCL12, (**k**) PCL15, (**l**) PCL20, (**m**) UF12 as a binder (the arrows indicate the binder presence).

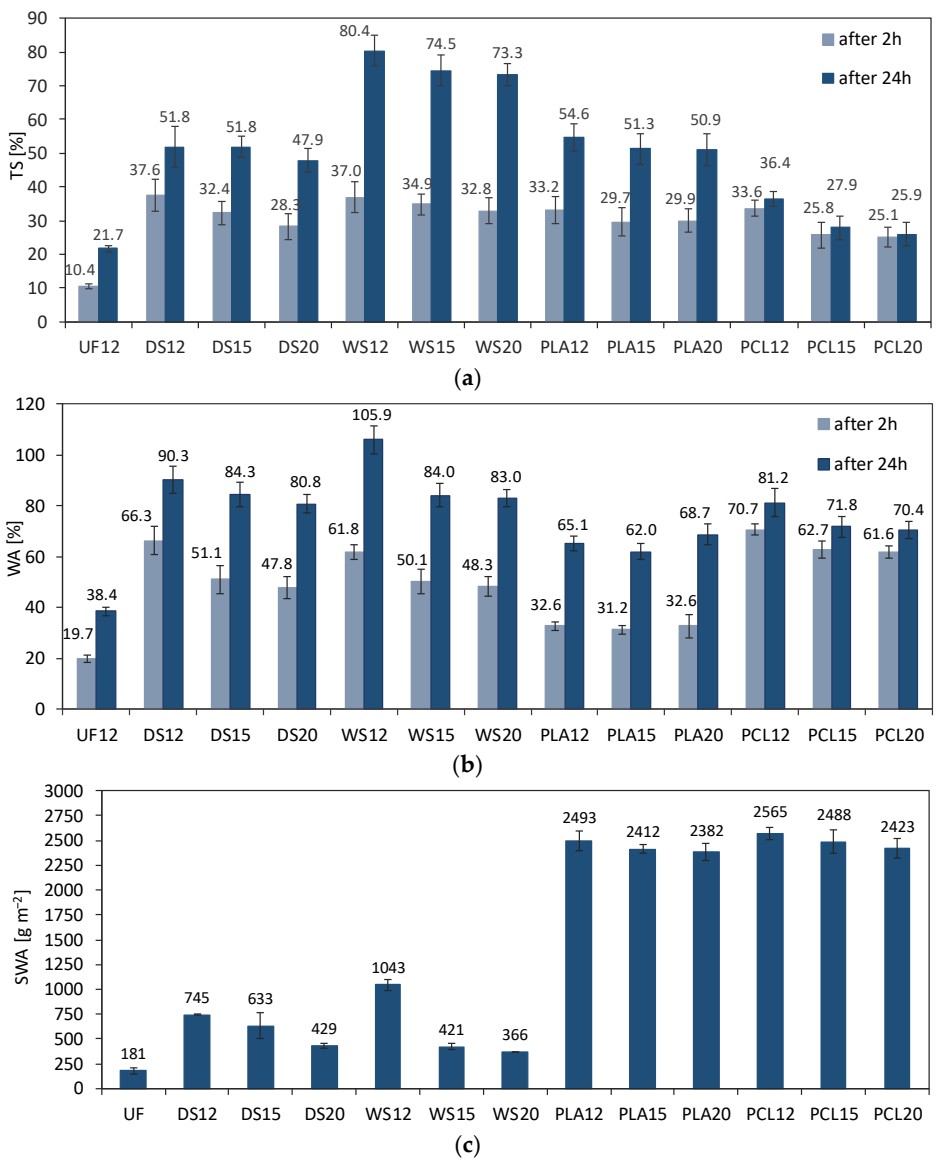

**Figure 7.** (**a**) Thickness swelling (TS), (**b**) water absorption (WA), and (**c**) surface water absorption (SWA) of tested HDF.

The results achieved for TS and WA are further proved in surface water absorption tests (SWA) (Figure 7c). The error bars in the graph represent the standard deviation. According to the results, the SWA decreases with the rise of the binder content. The lowest SWA average value was noted for UF12 (181 g m$^{-2}$) and the highest for PCL12 (2565 g m$^{-2}$) and PLA12 (2493 g m$^{-2}$). Significantly higher average values of SWA were found for the PLA and PCL samples. This may be the effect of the uneven spread of the binder over the fibers during blending, which causes the occurrence of local zones of binder-less fibers (Figure 6g–m). Despite the action of water on only the top surface for 2 h, PLA and PCL were impregnated with water over the entire thickness of the sample. Increasing the resination caused a slight decrease in the average SWA values. Reducing the resin content for PLA in extreme values from 12 to 20% resulted in a decrease in SWA by 4.6%, and for PCL by 5.5%. The statistical analysis shows that there are no statistically significant differences between the average values of the SWA results and the resination for the type of PLA and PCL binder used. The decreasing reaction to water of the panels made with the increasing amount of starch has been confirmed by [70].

Figure 8 presents the results of the contact angle measurements for all produced fiberboards. The error bars in the graph represent the standard deviation. The water sessile drop test conducted on DS and WS samples shows the highest hydrophobic behavior, possibly due to decreased porosity of the HDF with the binder with 20% resination. Therefore, as the resination increased for both 1 s and 60 s, the average values of the contact angle increased. For PLA and PCL, an increase in the contact angle in the range of resination from 12 to 15% was observed, both for 1 s and 60 s. The changes were highest for PCL20 (19.2% of reduction), PCL15, DS12, DS15, DS20—10.5%, 17.0%, 13.3%, 10.4%, respectively. The smallest changes in contact angle after 60 s were achieved for TPS: 1.9% for WS12, 1.8% for WS15, and 2.7% for WS20. A resination increase of 20% in HDF causes a decrease in the angle value for 1 s and 60 s, which means better surface wettability and higher surface energy for PLA20 and PCL20 [71,72]. The decrease in the wetting angle value was small; the increase in wettability from 15 to 20% for PLA resulted in a decrease in the contact angle by 1% after 1 s, while after 60 s the average value of the angle remained at the same level as for the 15% resin, at 102°. The PCL biopolymer caused a decrease after 1 s by 10%, and after 60 s by 7%. The increase in surface wettability with the increase in resination was probably caused by the uneven biopolymer coverage of the fibers (Figure 2). This is also explained by the analysis of SEM images, where voids and places where there is an excess of biopolymer are visible (Figure 6). Gumowska et al. [57] confirmed that with the increase in resination (12, 15, 20%) in particleboard bonded by PLA and PCL, the average values of the contact angle increased both after 1 s and after 60 s. The highest average contact angle after 1 s was recorded for DS12, DS15, DS20, reaching values of 112°, 113°, 115°, and respectively; the lowest after 1 s was recorded for PCL20, at 95°. The highest average contact angle after 60 s was recorded for UF12, WS12, WS15, and WS20, reaching values 105°, 105°, 107°, and 107°, respectively; the lowest after 60 s was recorded for PCL12, at 84°. For each tested HDF sample, after the water droplets remained on the surface for 60 s, the contact angle decreased.

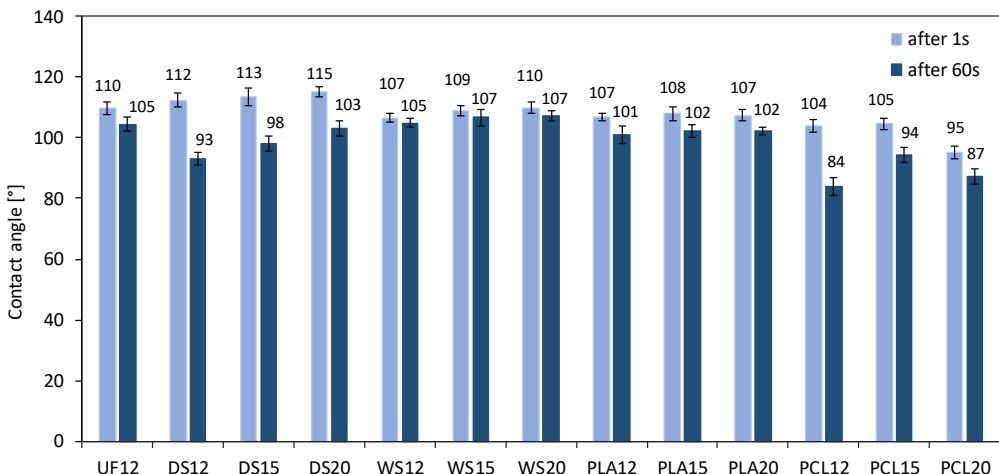

**Figure 8.** The contact angle of tested HDF.

## 4. Conclusions

On the basis of the completed research and analysis of the results, the following conclusions can be drawn:

A slightly higher densification of the face layer was found for panels where the water spray was applied over the mat surfaces before hot-pressing.

The bending features (MOR and MOE) depended on the binder type and resination level; there was increasing MOR and MOE with a rise in the DS resination; the maximum MOR and MOE values for WS were found for 15% resination; the bending properties of panels made with PLA and PCL decreased with the resination increase; however, the radical drop in MOR and MOE occurred when resination grew from 15% to 20%.

A similar relation between the binder type and resination level was found for IB; however, a significantly higher IB among the panels with biopolymers was registered for WS and PCL; the IB was not influenced by the density profile.

A slight, statistically insignificant reduction in TS, WA, and SWA was found for panels with increasing biopolymer binder content; an intensive increase in TS and WA was noted for WS after 24 h of soaking where the starch as a binder in wet form was mixed with fibers; the PLA panels had the lowest WA, and the lowest SWA among the biopolymer-bonded samples was for WS.

With the increase in resination for TPS from 12 to 20%, the value of the contact angle increased both after 1 s and 60 s. The increase in resination for PLA from 12 to 15% caused a statistically insignificant increase in the contact angle and a statistically insignificant decrease when resination grew from 15% to 20% for both 1 s and 60 s. The average value of the contact angle for PCL increased with the increase in resin content, while for PCL20 an increase in surface wettability was observed concerning PCL15.

In the case of dynamically evaporating solvents from binding mixtures, a development of fibers' resination (blending) techniques should be carried out, to avoid the uneven spreading of the binder over the resinated material. Further research on improving the strength of wood-based panels bonded with biopolymers should focus on improving their adhesion to the surface of wood/wood particles or fibers.

**Supplementary Materials:** The following are available online at https://www.mdpi.com/article/10.3390/f14010084/s1, Supplementary Material: Significant differences ($\alpha = 0.05$) between factors and levels.

**Author Contributions:** Conceptualization, A.G. and G.K.; methodology, A.G. and G.K.; formal analysis, A.G.; investigation, A.G.; resources, G.K.; writing—original draft preparation, A.G. and G.K.; writing—review and editing, A.G. and G.K.; visualization, A.G. All authors have read and agreed to the published version of the manuscript.

**Funding:** This research received no external funding.

**Data Availability Statement:** Not applicable.

**Conflicts of Interest:** The authors declare no conflict of interest.

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
