# Peer review of "Physical and Mechanical Properties of High-Density Fiberboard Bonded with Bio-Based Adhesives"

_forests, doi:10.3390/f14010084_

Round 1

Reviewer 1 Report

The manuscript deals with the investigation and evaluation of selected properties of high-density fiberboards, bonded with three different types of biobased adhesives, i.e. polylactic acid, polycaprolactone, and thermoplastic starch. In general, the manuscript is very well-written, structured and informative, but still needs some minor improvements before acceptance for publication in the Forests Journal. Please, see below my comments on your work:

In general, the title (lines 2-3), the abstract (lines 9 to 27) and the keywords (lines 28-29) correspond to the title, aims and objectives of the manuscript. The abstract is well-written and informative, and contains the main findings of the article.

In the title, I would suggest to replace “fibrous wood-based composites” with “high-density fiberboards" in order to be more precise.

Lines 17-18: “common use adhesive” should be “commonly used adhesive”, please revise.

Line 21: although well-known, please provide the full term, i.e. Scanning Electron Microscope, followed by the common abbreviation SEM.

Lines 39-40: the sentence is not very clear, please revise it. MDF is repeated twice.

Line 41: “hard board” should be “hardboards”, please revise.

Line 54: please remove www.verifiedmarketresearch.com , it is already included in the references. Data on the global market of wood adhesives as well as recent developments in the field of natural binders for wood composites can be found in this relevant recent reference: https://doi.org/10.1016/j.jmrt.2022.10.156

Line 55: please add the full terms, i.e. urea-formaldehyde and phenol-formaldehyde resins, followed by the abbreviations UF and PF.

Lines 93-95: speaking about alternative lignocellulosic feedstocks for development of wood composites, please check these relevant examples: https://doi.org/10.1016/j.jmrt.2022.08.166; https://doi.org/10.1016/j.indcrop.2021.114162; https://doi.org/10.1016/j.conbuildmat.2021.122906  

Lines 97-98: the examples of natural feedstocks for bio-based binders are relevant, but I’d suggest to add also tannins, vegetable oils, and natural rubber.

Lines 102-103: I think the company name (Nature Works) should be deleted, as this is a scientific article and not a commercial. The same comment applies to line 108 – please, remove the respective website from the main text of the manuscript.

Overall, the Introduction part is well written and informative, and provides relevant information and references on the research topic. However, I’d recommend to further elaborate it based on the comments given above.

Line 131: “industrial fibers” – please provide information/data about the supplier of the fibers.

Line 137: please provide information about the UF resin used, e.g. solids content, urea/formaldehyde ratio, viscosity, etc.

Line 150: “aimed density” – I believe “target density” is the more appropriate term.

Lines 153: please explain, justify the resination percentages used (12%, 15%, and 20%). Are they based on some preliminary trials, or literature data?

Lines 161-164: please explain/justify the pressing parameters used. In addition, please provide information on the hot press used (company producer, city, country).

Line 166: “Error! Reference source not found” – please check.

Line 170: please include information about Table 1 in the main text of the manuscript.

Lines 172-173: this sentence has already been given in lines 167-169, please check and revise.

Overall, the Materials and Methods section is well written and detailed, but can be further elaborated based on the comments above.

Lines 197-198: “Error! Reference source not found” – please check.

Line 223: “Error! Reference source not found” – please check.

Lines 237-238: “Error! Reference source not found” – please check. It should be Figure 3.

Lines 275-276: “Error! Reference source not found” – please check formatting. It should be Figure 4.

General comment on figures 3, 4, 7, and 8 – please indicate that the error bars represent the standard deviation.

In general, the results of the study are detailed, informative and properly discussed with relevant research works in the field.

The Conclusion part (lines 402-428) reflects the main findings of the manuscript. In addition to the practical application of your results, I’d recommend to add also the potential for future studies in the field.

The References cited are appropriate to the topic of the manuscript. Inclusion of additional references, especially in the Introduction and Results and Discussion section, will significantly increase the scientific merit of the presented manuscript. 

Reviewer 2 Report

General comments:

The article titled "Selected properties of fibrous wood-based composites bonded with biopolymers" presents insightful results in assessing the impact of natural bio-polymer binders thermoplastic starch-TPS, PLA, and PCL on selected physical and mechanical properties of HDF manufactured with different binders.

My general comments are:

·         The similarity index checking by Turnitin is around 24%, which is quite high for international publications. Authors should reduce it to below 20%.

·         The study is well documented, in line with the guidelines for the authors imposed by the journal.

·         The abstract is presented with too much background of the study. The brief quantitative results, discussion, and conclusion are missing.

·         The keywords are too many. Five important keywords are enough to help the manuscript in the journal's search engine.

·         The state-of-the-art written in the introduction of the paper is well documented and focused on the actual research direction in the field.

·         The introduction properly presents the aim of the study.

·         The experimental procedure is justified but less comprehensive.

·         The results are presented concisely and sustained by proper figures and tables.

·         Furthermore, compared to references, the results obtained are significant. The discussion section is clear and understandable and follows the results obtained.

·         The conclusion is well written to conclude the results.

·         The reference format should be revised according to the journal's requirements.

However, the article lacks a major scientific approach. Thus, the article can be accepted after addressing the below comments in the revision (major revision).

 Detail Comments:

1.       Title: The title is not written correctly. Please revise to "Physical and Mechanical Properties of High-Density Fiberboard Bonded with Bio-Based Adhesives."

2.       Abstract:

a.       Page 1. Line 9-14. Too much background of the study in the abstract. The brief quantitative results, discussion, and conclusion are missing. Please revise this.

b.       Page 1. Line 28-29. Five keywords are enough to help the manuscript in the journal's search engine.

3.       Introduction:

a.       The introduction is well-written. However, the authors failed to connect the published work on fibreboards bonded with bio-based adhesives. The following article could be used as references:

·         Tailoring of oxidized starch's adhesion using crosslinker and adhesion promotor for the recycling of fiberboards. J. Appl. Polym. Sci. 2019, 136, 47966. https://doi.org/10.1002/app.47966

·         Tuning of adhesion and disintegration of oxidized starch adhesives for the recycling of medium density fiberboard. Bioresources. 2020, https://doi.10.15376/biores.15.3.5156-5178

4.       Material and methods:

a.       Page 3. Section 2.2. The detailed preparation of each adhesive is missing. The authors should write the method in detail to ensure future repeatability.

b.       Page 3. Section 2.2. The basic properties of each adhesive are missing. The authors should write at least the solids content, the viscosity, and the gelation time of each adhesive.

c.       Page 4. Section 2.3. Line 159. Evaporation of solvent for 3 days? Is this necessary? The HDF is produced by hot pressing at 180°C for 10 min. The hot-pressing will evaporate the solvent.

d.       Page 4. Line 163. Why did the authors select 10 min for hot-pressing HDF with 3 mm thickness? The press factor is high.

e.       Page 4. Line 166. Reference error. Revise this.

f.        The authors should add the flow chart of HDF manufacturing with images of the process for better understanding.

g.       Page 4. Line 172-186. Please write a brief evaluation method of HDF properties, such as IB, MOR, MOE.

5.       Results and discussion:

a.       Many References error were detected. Revise this accordingly.

b.       The results are discussed very well. If possible, please add the results of FTIR and its discussion.

c.       The results of statistical analysis and the Duncan Post-hoc test are not discussed in the Results and Discussion. The authors should add this and discuss the significant result of each adhesive and which adhesive content (resination) is the optimum.

6.       Conclusions

a.       Please write which adhesive is suitable for HDF and its content (resination) according to the statistical 

Reviewer 3 Report

I have now completed my assessment of " Selected properties of fibrous wood-based composites bonded with biopolymers" for Forests. The manuscript is well-written and provides interesting information about the development of wood-based composites. However, I would reconsider the paper for publication after Major Revisions are made due to the following reasons.

(1). It is preferable to read the following papers: https://doi.org/10.1016/j.carbpol.2018.10.115, https://doi.org/10.1016/j.compositesb.2012.05.008, https://doi.org/10.1007/s13196-014-0111-5, 10.5897/AJB12.288 and compare the experimental approach, organization of the data and results and discussion of the present work with these similar types of papers. These articles can be read and cited in the Introduction and Results and Discussion sections to improve the manuscript’s quality.

(2). Is it cost-effective to use alternative binders or adhesives?

(3). The IB strength of the panel is often used to assess bonding efficacy. How do you explain the lower IB compared to the urea-formaldehyde adhesive?

(4) What are your strategies for reducing water absorption and thickness swelling?

Round 2

Reviewer 2 Report

The results of statistical analysis and the Duncan Post-hoc test are not discussed in the Results and Discussion. The authors did not provide the ANOVA tables and the Post-Hoc Test results. Please provide it briefly.

See the example:

·         Modification of Oxidized Starch Polymer with Nanoclay for Enhanced Adhesion and Free Formaldehyde Emission of Plywood. J. Polym. Environ. 2021, 29, 2993–3003. https://doi.org/10.1007/s10924-021-02101-w

Author Response

Dear Reviewer,
we collected all p-values we received after statistical evaluation of the results. Due to a large number of tested variants of panels (13), there is a significant number of data. So, we prefer to add the information in the manuscript (2.5. Statistical Analysis) that the detailed p-values have been attached as supplementary material.
Hope it will be ok for you.
With kind regards
on behalf of co-authors
Grzegorz Kowaluk

Reviewer 3 Report

In my point of view, this manuscript can be published in Polymers in its present form.

Author Response

Thank you so much for that positive opinion!